# Linearly Constrained Gaussian Processes are SkewGPs: application to Monotonic Preference Learning and Desirability

**Alessio Benavoli**[1]

**Dario Azzimonti**[2]

[1]School of Computer Science and Statistics, Trinity College Dublin, Ireland
[2]Dalle Molle Institute for Artificial Intelligence (IDSIA), USI/SUPSI, Lugano, Switzerland

## Abstract

We show that existing approaches to Linearly Constrained Gaussian Processes (LCGP) for regression, based on imposing constraints on a finite set of operational points, can be seen as Skew Gaussian Processes (SkewGPs). In particular, focusing on inequality constraints and building upon a recent unification of regression, classification, and preference learning through SkewGPs, we extend LCGP to handle monotonic preference learning and desirability, crucial for understanding and predicting human decision making. We demonstrate the efficacy of the proposed model on simulated and real data.

## 1 INTRODUCTION

Preference learning [18] aims at learning predictive preference models from data. Unlike regression or classification, where the target variable is a scalar, preference data is in the form of pairwise comparisons, which express a subject's preference between alternative options. Applications of preference learning are ubiquitous in recommendation systems across diverse domains, such as e-commerce, social media, and entertainment platforms. Consider a set $\mathscr{X} \subset \mathbb{R}^n$ and a binary relation $R$ on $\mathscr{X}$ expressed by a subject (that is $R$ is a subset of $\mathscr{X} \times \mathscr{X}$). Mathematically, a *strict preference*[1] is a binary relation, denoted by $\succ$, which is asymmetric and

negatively transitive [26, Ch. 2].[2] For instance, imagine you are planning a trip from destination A to destination B and have three train options. What is your preference?



Option 1: cost=5€, travel-time=15min,

Option 2: cost=7€, travel-time=10min,

Option 3: cost=3€, travel-time=20min.



In this case $\mathscr{X} = \mathbb{R}^2$ and $\mathbf{x}_1 = [5,15]^\top$, $\mathbf{x}_2 = [7,10]^\top$, $\mathbf{x}_3 = [3,20]^\top \in \mathscr{X}$. Stating that Option 1 is preferred to Option 2 is denoted as $\mathbf{x}_1 \succ \mathbf{x}_2$. Then, asymmetry implies that if $\mathbf{x}_1 \succ \mathbf{x}_2$ then $\mathbf{x}_2 \not\succ \mathbf{x}_1$. Negative transitivity implies that if $\mathbf{x}_1 \succ \mathbf{x}_2$ then either $\mathbf{x}_1 \succ \mathbf{x}_3$ or $\mathbf{x}_3 \succ \mathbf{x}_2$ or both. These are the minimum *consistency* properties defining a strict preference relation. However, in many applications, it is reasonable to assume further properties. For instance, in the above example, it seems natural to assume that any subject should prefer Option 5 to Option 4:



Option 4: cost=3€, travel-time=9min,

Option 5: cost=2€, travel-time=4min.



This property is called *strict monotonicity*: if $\mathbf{x}, \mathbf{y} \in \mathscr{X}$ and $\mathbf{x} \leq \mathbf{y}$, $\mathbf{x} \neq \mathbf{y}$ then $\mathbf{x} \succ \mathbf{y}$, (where $\leq$ means that any element of $\mathbf{x}$ is at least as small as the corresponding component $\mathbf{y}$).[3] Our objective is to learn strictly monotonic preferences from pairwise data. Assuming also continuity of the preference relation [26, Ch. 2], one can prove that any strictly monotonic preference is representable by a strictly monotone utility function $f$. Therefore, learning a preference can be formulated as the problem of learning monotonic utility functions that represents it.[4] In real-world scenarios, individuals, when expressing their preferences, often

---

[1]This paper focuses on strict preference rather than weak preference. Learning weak preference would result in a zero denominator in Bayes' rule when using continuous distributions like Gaussian Processes (GPs), as opposed to discrete distributions. In strict prefrence, we could incorporate a 'just noticeable difference' threshold to model situations, where a subject judges two options equivalent because the difference in their utility is small (below a threshold).

[2]*Asymmetric*: $\forall \mathbf{x}, \mathbf{y} \in \mathscr{X}$ if $\mathbf{x}R\mathbf{y}$ then not $\mathbf{y}R\mathbf{x}$. *Negatively transitive*: if $\mathbf{x}R\mathbf{y}$ then for any other element $\mathbf{z} \in \mathscr{X}$ either $\mathbf{x}R\mathbf{z}$ or $\mathbf{z}R\mathbf{y}$ or both.

[3]Depending on the application, we can obviously define strict monotonicity by changing the direction $\mathbf{x} \geq \mathbf{y}$, $\mathbf{x} \neq \mathbf{y}$ then $\mathbf{x} \succ \mathbf{y}$.

[4]This representation is not unique. Utility functions are invariant under increasing transformations. We can define a new utility function $g(u(x))$ for any increasing function $g$, that is

deviate from these consistencies properties for different reasons. Accurately representing erroneous preferences requires modellings errors through tools like random utility models [33, 34], i.e., a subject's preference is determined by a noisy utility function. This is crucial for learning because it requires us to define a likelihood function for preference data that accounts for these errors.

A powerful way to learn unknown functions is through *Gaussian Processes* (GP)[40, 48], which are priors over functions. A GP-based method to learn preference learning was firstly proposed by [14, 25] with a *probit* likelihood (to account for errors).[5] This approach offers two advantages: a nonlinear utility in the covariates and the representation of uncertainty through the posterior. Since the posterior is not Gaussian, Chu and Ghahramani [14] proposed the Laplace's approximation for inference. Other approximations were considered in [25]. More recently, [9, 8] showed that the posterior has a closed-form, called SkewGP, and exploited this relationship to efficiently sample from the posterior. Applications of preference learning for active learning and Bayesian optimisation, have been investigated by [50, 23, 52, 10, 39, 11].

Many recent works [49, 58, 1, 15, 22, 30, 29, 28, 31, 27, 3] developed GP models for regression that satisfy monotonic, or more generally linear inequality constraints. In particular, [49, 58, 1, 15, 22] enforce monotonicity constraints by imposing them on a finite set of operational points. The works in [30, 29, 28, 31] exploit a finite-dimensional kernel to extend the monotonicity constraint to the whole domain. Other approaches [27, 3] impose shape constraints through squared Gaussian process derivatives and series expansions. Finally *monotonic-GP-flow* [55], imposes monotonicity on GPs based on numerical solutions of stochastic differential equations. We point the reader to [53] for a comprehensive survey study of LCGP, including bound constraints, monotonicity and linear partial differential operator constraints. In this work we aim to bring the recent advances in modelling monotonicity constraints with GPs for regression to the preference learning setting.

The contributions of this work are the following:

- We show that linearly constrained Gaussian Processes (LCGP) that impose monotonicity constraints with a finite set of operational points are SkewGPs.

$x \succ y$ *iff* $u(x) > u(y)$ *iff* $g(u(x)) > g(u(y))$. So $u$ and $g(u)$ represent the same strict preference relation. We will go back to this point later in in the paper.

[5]Due the probit likelihood, in the preference $x \succ y$ iff $u(x) > u(y)$, now also the magnitude of their utility difference $u(x) - u(y)$ is important (and not only the relative ranking of two items $u(x) > u(y)$). The reason is that we are assuming that the probability of errors is proportional to $|u(x) - u(y)|$, that is it is more difficult for a subject to express a preference between two options with close utility. This makes the scale of the utility function to be identifiable.

- Exploiting the conjugacy of SkewGPs with the normal and probit likelihood (and their product) [8], we extend LCGP models to preference learning and classification tasks, deriving a novel nonparametric model for monotonic preference learning and desirability learning (which is equivalent to a monotonic classification problem [37, 13]).
- We compare our SkewGP-formulation of monotonic regression and preference learning against *monotonic-GP-flow* on 7 1D benchmark functions. Our SkewGP outperforms monotonic-GP-flow in both accuracy and uncertainty quantification.
- We apply SkewGP to two preference datasets demonstrating that models without monotonicity constraints can produce wrong predictions, thus highlighting the importance of incorporating monotonicity constraints.

In this work, we focus on monotonicity constraints, we leave the extension to any linear inequality constraint for future work.

## 2 SKEW-NORMAL DISTRIBUTION AND SKEW-GAUSSIAN PROCESSES

The unified skew-normal distribution [4, 6, 17, 2] generalises the normal distribution by allowing for non-zero skewness. A vector $\mathbf{z}$ distributed as skew-normal can be constructed from a multivariate normal which is truncated in part of its component, see [6, Ch.7]. Consider two vectors $\mathbf{z}_0 \in \mathbb{R}^s, \mathbf{z}_1 \in \mathbb{R}^p$ such that:

$$\begin{bmatrix} \mathbf{z}_1 \\ \mathbf{z}_0 \end{bmatrix} \sim N(\mathbf{0}_{s+p}, M), \ M = \begin{bmatrix} \Omega & \Delta \\ \Delta^\top & \Gamma \end{bmatrix}, \tag{1}$$

where $M$ is a full-rank covariance matrix. Define $\zeta$ to be distributed as $N(\mathbf{z}_1 | \mathbf{z}_0 + \gamma > \mathbf{0}_s)$, where $\gamma \in \mathbb{R}^s$ and the inequality $\mathbf{z}_0 + \gamma > \mathbf{0}_s$ holds component-wise.[6] Then, given a location vector $\xi \in \mathbb{R}^p$, $\mathbf{z} = \xi + \zeta \in \mathbb{R}^p$ is distributed as a *multivariate unified skew-normal distribution* with latent skewness dimension $s$. We denote $\mathbf{z} \sim \text{SUN}_{p,s}(\xi, \Omega, \Delta, \gamma, \Gamma)$ and its Probability Density Function (PDF) is given by:

$$p(\mathbf{z}) = \phi_p(\mathbf{z} - \xi; \Omega)$$
$$\frac{\Phi_s\left(\gamma + \Delta^\top \Omega^{-1}(\mathbf{z} - \xi); \Gamma - \Delta^\top \Omega^{-1}\Delta\right)}{\Phi_s(\gamma; \Gamma)}, \tag{2}$$

where $\phi_p(\mathbf{z} - \xi; \Omega)$ denotes the PDF of a multivariate normal distribution with mean $\xi \in \mathbb{R}^p$ and covariance $\Omega \in \mathbb{R}^{p \times p}$. $\Phi_s(\mathbf{a}; M)$ represents the Cumulative Distribution Function (CDF) of a $s$-dimensional multivariate normal distribution with zero mean and covariance matrix $M$ evaluated at $\mathbf{a} \in \mathbb{R}^s$. The parameters $\gamma \in \mathbb{R}^s, \Gamma \in \mathbb{R}^{s \times s}, \Delta \in \mathbb{R}^{p \times s}$

[6]Note that, in the standard construction of the SUN distribution [6, Ch.7], the matrix $M$ is a correlation matrix. However, we can obtain the standard construction from (1) by a change of variables.

control the skewness of the distribution, in particular $\Delta$ is called *skewness matrix*. When $\Delta = 0$, eq. (2) reduces to $\phi_p(\mathbf{z} - \boldsymbol{\xi}; \Omega)$, i.e. a skew-normal with zero skewness matrix is a normal distribution. Moreover, we assume that $\Phi_0(\cdot) = 1$, so that, for $s = 0$, eq. (2) becomes a multivariate normal distribution. Figure 1 shows the density of a univariate SUN distribution with latent dimensions $s = 1$ and $s = 2$.

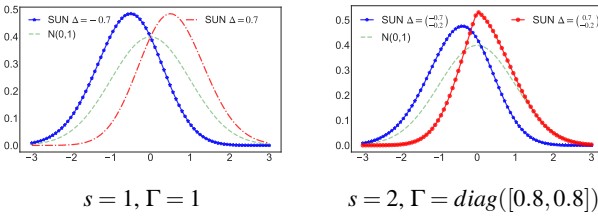

$$s = 1, \Gamma = 1 \qquad\qquad s = 2, \Gamma = diag([0.8, 0.8])$$

Figure 1: Density plots for $SUN_{1,s}(0, 1, \Delta, \gamma, \Gamma)$. For all plots $\Gamma$ is a correlation matrix, $\gamma = 0$, dashed lines are the contour plots of $y \sim N_1(0, 1)$.

A SkewGP [7, 8] is a generalisation of a SUN distribution to a stochastic process which becomes a GP when skewness is zero, see Appendix A for a quick recap on GPs. To define a SkewGP, we consider here a location function $\xi : \mathbb{R}^d \to \mathbb{R}$, a scale (kernel) function $\Omega : \mathbb{R}^d \times \mathbb{R}^d \to \mathbb{R}$, a skewness vector function $\Delta : \mathbb{R}^d \to \mathbb{R}^s$ and the parameters $\boldsymbol{\gamma} \in \mathbb{R}^s, \Gamma \in \mathbb{R}^{s \times s}$. A real function $f : \mathbb{R}^d \to \mathbb{R}$ is SkewGP-distributed with latent dimension $s$, if for any sequence of $n$ points $\mathbf{x}_1, \dots, \mathbf{x}_n \in \mathbb{R}^d$, the vector $[f(\mathbf{x}_1), \dots, f(\mathbf{x}_n)]^\top \in \mathbb{R}^n$ is SUN distributed with parameters $\boldsymbol{\gamma}, \Gamma$ and location, scale and skewness matrices, respectively, given by

$$
\begin{aligned}
\xi(X) &= \left[\xi(\mathbf{x}_1), \xi(\mathbf{x}_2), \dots, \xi(\mathbf{x}_n)\right]^\top, \\
\Omega(X, X) &= \begin{bmatrix} \Omega(\mathbf{x}_1, \mathbf{x}_1) & \Omega(\mathbf{x}_1, \mathbf{x}_2) & \dots & \Omega(\mathbf{x}_1, \mathbf{x}_n) \\ \Omega(\mathbf{x}_2, \mathbf{x}_1) & \Omega(\mathbf{x}_2, \mathbf{x}_2) & \dots & \Omega(\mathbf{x}_2, \mathbf{x}_n) \\ \vdots & \vdots & \dots & \vdots \\ \Omega(\mathbf{x}_n, \mathbf{x}_1) & \Omega(\mathbf{x}_n, \mathbf{x}_2) & \dots & \Omega(\mathbf{x}_n, \mathbf{x}_n) \end{bmatrix}, \\
\Delta(X) &= \left[\Delta(\mathbf{x}_1), \Delta(\mathbf{x}_2), \dots, \Delta(\mathbf{x}_n)\right],
\end{aligned}
\tag{3}
$$

where $X = [\mathbf{x}_1, \mathbf{x}_2, \dots, \mathbf{x}_n]^\top$. In this case, we write $f(\mathbf{x}) \sim SkewGP_s(\xi(\mathbf{x}), \Omega(\mathbf{x}, \mathbf{x}), \Delta(\mathbf{x}), \boldsymbol{\gamma}, \Gamma)$. SkewGPs are conjugate with both the normal and affine probit likelihood and, more in general, with their product. This allows us to derive their posterior for nonparametric regression, classification, preference learning and mixed problems.

In particular, consider the affine-probit-normal product likelihood:

$$
\begin{aligned}
p(Y, Z, W \mid f(X)) &= \Phi_{m_a}(Z + W f(X); \Sigma) \\
&\quad \cdot \phi_{m_r}(Y - C f(X); R).
\end{aligned}
\tag{4}
$$

where $m_r$ (the subscript $r$ stands for regression) denotes the number of regression-type observations and $m_a$ the number of binary/preference-type observations (the subscript $a$ stands for affine). Therefore, we have that $Y \in$

$\mathbb{R}^{m_r}$, $C \in \mathbb{R}^{m_r \times n}$, $W \in \mathbb{R}^{m_a \times n}$, $Z \in \mathbb{R}^{m_a \times 1}$. The matrices $\Sigma \in \mathbb{R}^{m_a \times m_a}$, $R \in \mathbb{R}^{m_r \times m_r}$ are covariance matrices. This likelihood encompasses all the standard likelihood functions used in regression, classification and preference-learning. For instance, a standard regression is obtained by setting $C = I_{m_r}$, $R = \sigma^2 I_{m_r}$ and $m_a = 0$; classification is obtained for $W = diag(2Y - 1)$, $Z = \mathbf{0}_{m_a}$, $\Sigma = I_{m_a}$ and $m_r = 0$, where $Y$ is the vector containing the observed class values $Y_i \in \{0, 1\}$. Preference learning is obtained with $Z = \mathbf{0}_{m_a}$, $\Sigma = I_{m_a}$ and $m_r = 0$ and $W \in \mathbb{R}^{m_a \times n}$ whose s-th row is all zero apart from $W_i = 1, W_j = -1$ if the data includes the preference $\mathbf{x}_i \succ \mathbf{x}_j$.

We now report this result from [8, Theorem 3].

**Proposition 1.** *Let us assume a SkewGP prior* $f(\mathbf{x}) \sim SkewGP_s(\xi(\mathbf{x}), \Omega(\mathbf{x}, \mathbf{x}), \Delta(\mathbf{x}), \boldsymbol{\gamma}, \Gamma)$, *the likelihood* (4), *then a-posteriori* $f(\mathbf{x})$ *is SkewGP with mean, covariance and skewness functions:*

$$
\begin{aligned}
\tilde{\boldsymbol{\xi}}(\mathbf{x}) &= \boldsymbol{\xi}(\mathbf{x}) \\
&\quad + \Omega(\mathbf{x}, X) C^T (C\Omega(X, X) C^T + R)^{-1} (Y - C\xi(X)), \\
\tilde{\Omega}(\mathbf{x}, \mathbf{x}) &= \Omega(\mathbf{x}, \mathbf{x}) \\
&\quad - \Omega(\mathbf{x}, X) C^T (C\Omega(X, X) C^T + R)^{-1} C\Omega(X, \mathbf{x}), \\
\tilde{\Delta}(\mathbf{x}) &= \begin{bmatrix} \Delta(\mathbf{x}) & \Omega(\mathbf{x}, X) W^T \end{bmatrix} \\
&\quad - \Omega(\mathbf{x}, X) C^T (C\Omega(X, X) C^T + R)^{-1} C \\
&\quad \cdot \begin{bmatrix} \Delta(X) & \Omega(X, X) W^T \end{bmatrix}, \\
\tilde{\boldsymbol{\gamma}} &= \boldsymbol{\gamma}_p + \begin{bmatrix} \Delta(X) & \Omega(X, X) W^T \end{bmatrix}^T \\
&\quad \cdot \Omega(X, X)^{-1} (\tilde{\boldsymbol{\xi}}(X) - \boldsymbol{\xi}(X)) \\
\tilde{\Gamma} &= \Gamma_p - \begin{bmatrix} \Delta(X) & \Omega(X, X) W^T \end{bmatrix}^T \\
&\quad \Omega^{-1}(X, X) \begin{bmatrix} \Delta(X) & \Omega(X, X) W^T \end{bmatrix} \\
&\quad + \Delta_p^T \tilde{\Omega}(X, X)^{-1} \Delta_p, \\
\Delta_p &= \tilde{\Omega}(X, X) \Omega^{-1}(X, X) \\
&\quad \cdot \begin{bmatrix} \Delta(X) & \Omega(X, X) W^T \end{bmatrix}, \\
\boldsymbol{\gamma}_p &= [\boldsymbol{\gamma}, \ Z + W\xi(X)]^T, \\
\Gamma_p &= \begin{bmatrix} \Gamma & \Delta(X)^T W^T \\ W\Delta(X) & (W\Omega(X, X) W^T + \Sigma) \end{bmatrix}.
\end{aligned}
$$

The computation of predictive inference (posterior mean, credible intervals etc.) can be achieved by sampling the posterior SkewGP. Recall [6, Ch.7] that $\mathbf{z} \sim SUN_{p,s}(\boldsymbol{\xi}, \Omega, \Delta, \boldsymbol{\gamma}, \Gamma)$ can be written as $\mathbf{z} = \boldsymbol{\xi} + \mathbf{r}_0 + \Delta\Gamma^{-1}\mathbf{r}_{1,-\gamma}$ with $\mathbf{r}_0 \sim \phi_p(0; \bar{\Omega} - \Delta\Gamma^{-1}\Delta^T)$ and $\mathbf{r}_{1,-\gamma}$ is the truncation below $\gamma$ of $\mathbf{r}_1 \sim \phi_s(0; \Gamma)$. Note that sampling $\mathbf{r}_0$ can be achieved efficiently with standard methods, and $\mathbf{r}_{1,-\gamma}$ can be obtained efficiently using methods such as Gibbs sampling [54] linear elliptical slice sampler [20], minimax tilting method accept-reject sampler [12] and Hamiltonian Monte-Carlo [42].

Similarly to GPs, the functions and matrices defining a SkewGP, $SkewGP_s(\xi(\mathbf{x}), \Omega(\mathbf{x}, \mathbf{x}), \Delta(\mathbf{x}), \boldsymbol{\gamma}, \Gamma)$ may depend

on hyperparameters $\boldsymbol{\theta}$. These parameters are chosen by maximising the marginal likelihood, which is equal to:

$$p(Y) = \phi_{m_r}(Y - C\boldsymbol{\xi}(X); C\Omega(X,X)C^T + R)\frac{\Phi_{s+m_a}(\tilde{\boldsymbol{\gamma}}; \tilde{\Gamma})}{\Phi_s(\boldsymbol{\gamma}; \Gamma)},$$
(5)

with $\tilde{\boldsymbol{\gamma}}, \tilde{\Gamma}$ are defined in Proposition 1. This involves the computation of a high-dimensional multivariate CDFs $\Phi_{s+m_a}(\cdot), \Phi_s(\cdot)$. We use a variational inference technique to approximate the posterior distribution with a Gaussian distribution. This provides a lower bound for (5), which we maximise to find the hyperparameters.

## 3 A LINEARLY CONSTRAINED GP IS A SKEWGP

We recall [48, Sec. 9.4] that if $f : \mathbb{R}^D \to \mathbb{R}$ is GP distributed, that is $f \sim \mathrm{GP}(0,k)$ with kernel $k$, then its first derivative $\frac{\partial f_i}{\partial x_{ik}}$ is also GP-distributed with covariance (kernel):

$$k^I(\mathbf{x}_i, \mathbf{x}_j) := \mathrm{cov}\left(f_i, \frac{\partial f_j}{\partial x_{jl}}\right) = \frac{\partial k(\mathbf{x}_i, \mathbf{x}_j)}{\partial x_{jl}},$$
(6)

$$k^{II}(\mathbf{x}_i, \mathbf{x}_j) := \mathrm{cov}\left(\frac{\partial f_i}{\partial x_{il}}, \frac{\partial f_j}{\partial x_{je}}\right) = \frac{\partial k(\mathbf{x}_i, \mathbf{x}_j)}{\partial x_{il}\partial x_{je}},$$
(7)

for each $i, j, l, e \in \{1, 2, \ldots, D\}$.

We introduce a vector $U = [\mathbf{u}_1, \mathbf{u}_2, \ldots, \mathbf{u}_r]^\top$, with $\mathbf{u}_i \in \mathbb{R}^D$, of *operational points* and define $\mathbf{f}'(\mathbf{u}_i) = [\frac{\partial}{\partial u_{i1}}f(\mathbf{u}_i), \ldots, \frac{\partial}{\partial u_{iD}}f(\mathbf{u}_i)]^\top$. We assume that the vector $[f(\mathbf{x}_1), \ldots, f(\mathbf{x}_n), f(\mathbf{u}_1), \ldots, f(\mathbf{u}_r), \mathbf{f}'(\mathbf{u}_1)^\top, \ldots, \mathbf{f}'(\mathbf{u}_r)^\top]^\top$ is GP distributed with zero-mean and covariance matrix

$$M = \begin{bmatrix} K(X,X) & K(X,U) & K^I(X,U) \\ K(X,U)^\top & K(U,U) & K^I(U,U) \\ K^I(X,U)^\top & K^I(U,U) & K^{II}(U,U) \end{bmatrix}.$$
(8)

We define a linearly constrained GP by imposing:

$$L\begin{bmatrix} f(\mathbf{u}_1) \\ \vdots \\ f(\mathbf{u}_r) \\ \mathbf{f}'(\mathbf{u}_1) \\ \vdots \\ \mathbf{f}'(\mathbf{u}_r) \end{bmatrix} + \boldsymbol{\gamma} > 0.$$
(9)

It is immediate to verify that, by suitably selecting $L, \boldsymbol{\gamma}$, (9) allows us to impose bound and monotonicity constraints on $f$ at $U$. We could similarly impose constraints on the second derivative, integral of $f$ or on other affine operators, which preserve Gaussianity.

**Theorem 1.** *Assume that* $[f(\mathbf{x}), \mathbf{f}(\mathbf{u}_1), \ldots, \mathbf{f}(\mathbf{u}_r), \mathbf{f}'(\mathbf{u}_1)^\top, \ldots, \mathbf{f}'(\mathbf{u}_r)^\top]^\top$ *is GP distributed with zero-mean and covariance matrix* (8). *Then,*

*subject to the constraint* (9), $f(\mathbf{x}) + \boldsymbol{\xi}(\mathbf{x})$ *is SkewGP distributed with parameters* $\boldsymbol{\gamma}, \boldsymbol{\xi}(\mathbf{x})$,

$$\Gamma = \begin{bmatrix} LK(U,U)L^T & LK^I(U,U)L^\top \\ LK^I(U,U)L^\top & LK^{II}(U,U)L^\top \end{bmatrix},$$
(10)

$$\Delta(\mathbf{x}) = \begin{bmatrix} K(\mathbf{x},U)L^\top & K^I(\mathbf{x},U)L^\top \end{bmatrix},$$
(11)

*and scale function* $\Omega(\mathbf{x},\mathbf{x}) = k(\mathbf{x},\mathbf{x})$.

The proof of this and next theorems is in Appendix B. This result allows us to leverage the SkewGP as a prior distribution over functions and compute posteriors for regression, classification, and preference learning tasks (as shown in Proposition 1). These posteriors are guaranteed to satisfy the specified linear constraint at all operational points. In the next section, we will illustrate this theorem with a concrete example related to monotonicity.

## 4 MONOTONIC GP

In this section, we demonstrate how SkewGPs offer a unified approach to imposing monotonicity constraints on GPs. We achieve this by showing how SkewGPs can encompass various existing methods from the literature. This unification, combined with the conjugacy property of SkewGPs established in Proposition 1 and Theorem 1, enables us to extend these approaches beyond regression, which has been the primary focus of previous work on linearly constrained GPs. As a simple illustration of imposing monotonicity constraints in regression and preference learning, we will consider the function $f(x) = 3/(1 + \exp(-20x + 10))$ for $x \in [0,1]$ shown in Figure 2. We generated

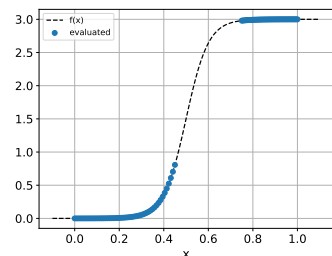

Figure 2

a regression and a preference learning dataset as follows. We first evaluated $f$ at 50 equally spaced points in $[0,0.45]$ and at 50 equally spaced points in $[0.75,1]$. The 100 points $\{x_l\}_{l=1}^{100}$ were used to generate regression data with $y_i = f(x_i) + \varepsilon_i$ where $\varepsilon_i \sim N(0, 0.0225)$. From $\{x_l\}_{l=1}^{100}$ we also generated 200 pairwise preferences as follows: $x_i \succ x_j$ if $f(x_i) + \varepsilon_i > f(x_j) + \varepsilon_j$ with $\varepsilon_i, \varepsilon_j \sim N(0, 0.0225)$, for randomly selected $x_i, x_j \in \{x_l\}_{l=1}^{100}$.[7]

### 4.1 INFINITE-DIMENSIONAL KERNEL

Existing approaches [49, 58, 1, 15, 22] enforce monotonicity constraints by holding them at specific operational points,

---

[7] $\int I_{f(x_i)+\varepsilon_i > f(x_j)+\varepsilon_j}(\varepsilon_i, \varepsilon_j)dN(\varepsilon_i, 0, \sigma^2)dN(\varepsilon_j, 0, \sigma^2)$ $=$ $\Phi_1(\frac{f(x_i)-f(x_j)}{\sqrt{2}\sigma})$, which gives rise to the probit likelihood in preference learning.

denoted as $U$. These approaches can be applied with any kernel, including infinite-dimensional ones, but in general, they guarantee global monotonicity only with high probability. Specific methods for imposing constraints and inference vary across these works. For a detailed comparison, please refer to Agrell [1, Table 1]. Our SkewGP-based approach incorporates methods like [58, 15] that perfectly enforce monotonicity constraints at operational points. Note that, for both regression and preference learning, all computations are performed analytically as described in Proposition 1. By leveraging the analytical derivations, we efficiently obtain the posterior samples (and so mean and credible region) using tailored MCMC methods as described in Section 2. This translates to fast inference. It is worth noting that methods using soft constraints with indicators replaced by probit function (or Normal likelihood) can also be formulated as SkewGP with different parameters, which will reduce to the frameworks in [49, 1, 22].

In order to apply these methods with SkewGPs therefore we only need to define $k, k^I, k^{II}$. For instance, the $D$-dimensional squared-exponential (SE) kernel is

$$k(\mathbf{x}_i, \mathbf{x}_j) = \sigma_0^2 \exp\left(-\sum_{d=1}^{D} \frac{(x_{id} - x_{jd})^2}{2\ell_d^2}\right),$$

$$k^I(\mathbf{x}_i, \mathbf{x}_j) = -\sigma_0^2 \exp\left(-\sum_{d=1}^{D} \frac{(x_{id} - x_{jd})^2}{2\ell_d^2}\right)\ell_l^{-2}(x_{il} - x_{jl}),$$

$$k^{II}(\mathbf{x}_i, \mathbf{x}_j) = \sigma_0^2 \exp\left(-\sum_{d=1}^{D} \frac{(x_{id} - x_{jd})^2}{2\ell_d^2}\right)$$
$$\ell_l^{-2}\left(\delta_{lh} - \ell_h^{-2}(x_{il} - x_{jl})(x_{ih} - x_{jh})\right),$$

respectively, where $\delta_{lh} = 1$ if $l = h$ and 0 otherwise and $\sigma_0, \ell_d$ for $d = 1, \ldots, D$ are the hyperparameters of the kernel. Figure 3 reports the sampled posterior SkewGP for both regression and preference learning using the dataset generated from the function in Figure 2. We used a SE kernel with $\ell = 0.15$, $\sigma_0 = 1$ for regression and $\sigma_0 = 90$ for preference and imposed the constraints on equally spaced operational points. Note that this approach can be applied directly to multi-dimensional functions. Figure 3 shows that the posterior inference improves in the constrained case (given the original function $f$ is monotonic) compared to the unconstrained case. In the constrained case, however, the samples do not preserve monotonicity globally. This is a known drawback of the approaches [49, 58, 1, 15, 22]. Several techniques exist for selecting the location of operational points $U$. We refer to [1] for a review of these techniques. Appendix C details how they are selected in the experimental section.

## 4.2 FINITE-DIMENSIONAL KERNEL

A way to impose constraints in the whole domain was proposed by [30, 29, 28, 31]. They achieve this through finite-

dimensional approximations of the GP that converge uniformly at the increase of the number of the knots. Here we follow [28] and consider degree 2 monotone splines (M-spline, [47]). To define a M-spline of degree 2, we consider $l + 1$ grid points (knots) $(t_0, \ldots, t_{l+1})$ such that $t_0 < t_1 < \cdots < t_l < t_{l+1}$. M-spline are piecewise polynomials defined as:

$$M_i(x) = \begin{cases} \frac{x - t_{i-1}}{t_i - t_{i-1}} & t_{i-1} \le x \le t_i \\ \frac{t_{i+1} - x}{t_{i+1} - t_i} & t_i \le x \le t_{i+1} \end{cases} \qquad (12)$$

for $i = 1, \ldots, l$. Figure 4 shows the polynomial for $l = 8$ and $\{t_i\}_{i=1}^{\ell}$ equally spaced in $[0,1]$ (and $t_0 = -1, t_{\ell+1} = 2$).

Then, the finite-dimensional GP is defined as

$$f(x) = \sum_{i=1}^{l} \beta_i M_i(x),$$
$$(13)$$

where $\beta_i$ are Gaussian distributed with zero-mean and covariance matrix $E[\beta_i, \beta_j] = \check{k}(t_i, t_j)$, where $\check{k}$

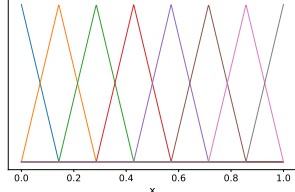

Figure 4: M-spline.

is a kernel. In the rest of the paper, we assume that $\check{k}$ is the SE kernel. It is then immediate to verify that $f$ is GP distributed with zero-mean and covariance kernel

$$k(x, x') = \sum_{i=1}^{l} \sum_{j=1}^{l} M_i(x)\check{k}(t_i, t_j)M_j(x'). \qquad (14)$$

We call $k$ in eq. (14) the 'MSP' kernel. We now show how we can impose monotonicity using SkewGP.

**Theorem 2.** *Consider $l$ operational points $[u_1, \ldots, u_{l+1}]$ defined as $u_i = (t_i + t_{i-1})/2$, then the SkewGP obtained from Theorem 1 with $L = diag([\mathbf{0}_r, I_r])$ and kernel defined as in (14) is monotone increasing in $[t_1, t_l]$.*

Therefore, we can also include the approaches [30, 29, 28, 31] into the SkewGP framework exploiting Proposition 1 ad Theorem 1. As before, we only need to compute $k^I(x, x'), k^{II}(x, x')$. Figure 5 shows the posterior SkewGP obtained with the kernel (14). Compared to Figure 3, it can be noticed that the mean and trajectories are piecewise linear and, more importantly, the monotonicity constraint holds globally in the interval $[0, 1]$, i.e., all sampled trajectories are monotonic in $[0, 1]$.

In [28], the extension to the multidimensional case $\mathbf{x} = [x_1, \ldots, x_D]^\top \in \mathbb{R}^D$ is obtained by considering an additive model $f(\mathbf{x}) = \sum_{d=1}^{D} f(x_d)$ and, therefore, an additive kernel $k(\mathbf{x}, \mathbf{x}') = \sum_{d=1}^{D} \sum_{i=1}^{l} \sum_{j=1}^{l} M_{di}(x_d)\check{k}(t_{di}, t_{dj})M_{dj}(x'_d)$. This is the approach we will follow in the rest of the paper. Note that, it is also possible to use the product kernel, similar to [30] or the ANOVA kernel (including both sums and products). The additive kernel holds the advantage of scaling more effectively to high dimensions.

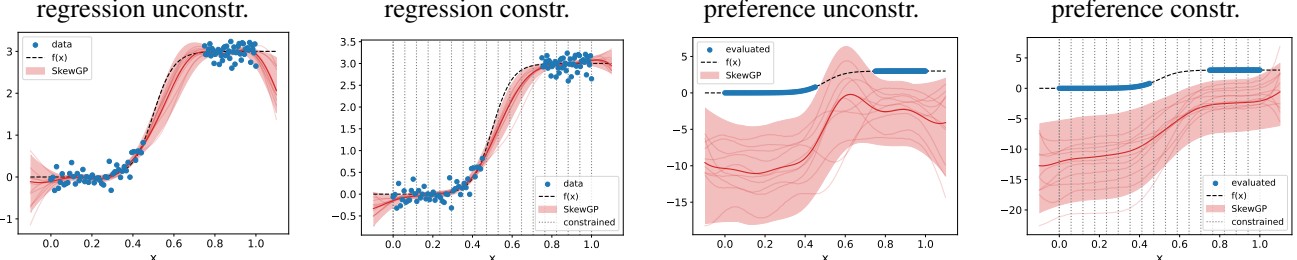

Figure 3: SkewGP with RBF kernel, $\ell = 0.15$, $\sigma = 1$ for regression and $\sigma = 90$ for preference. The thick red line shows the posterior mean, and the shaded region represents the 95% credible interval. Ten sampled functions are also included to illustrate the uncertainty. Vertical lines denote the operational points where the monotonicity constraint is enforced.

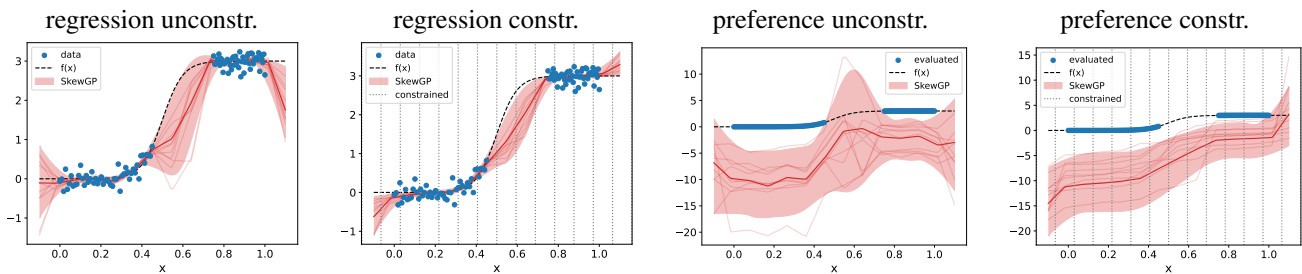

Figure 5: SkewGP with MSP kernel, $\ell = 0.1$, $\sigma = 1$ for regression and $\sigma = 90$ for preference. The thick red line shows the posterior mean, and the shaded region represents the 95% credible interval. Ten sampled functions are also included to illustrate the uncertainty. Vertical lines denote the operational points where the monotonicity constraint is enforced.

## 4.3 TRANSFORMED GP

The works [27, 3] designed methods for imposing shape constraints on functions through squared Gaussian process derivatives and basis expansions. In particular, they approximate the kernel with a basis expansion $k(x,x') \approx Cov(h(x), h(x'))$ with $h(x) = \sum_{i=1}^{m} \beta_i \phi_i(x)$ where $\beta_i$ are independent Gaussian distributed variables and $\phi_i$ are basis functions derived from the eigenfunctions of the Laplace operator. Then they build a monotonic function as

$$h^+(x) = \int_{-\infty}^{x} \left( \sum_{i=1}^{m} \beta_i \phi_i(z) \right)^2 dz, \quad (15)$$

which is equal to $h^+(x) = \sum_{i=1}^{m} \sum_{j=1}^{m} \beta_i \beta_j \int_{-\infty}^{x} \phi_i(z) \phi_j(z) dz$. Note the nonlinearity introduced by the multiplication between the coefficients $\beta_i$. This breaks the connection with SkewGP and, therefore, the conjugacy with normal and probit-affine likelihoods. Moreover, the basis function $\int_{-\infty}^{x} \phi_i(z) \phi_j(z) dz$ loses interpretability. We will show next that we can build on the same ideas proposed in [27, 3], while preserving linearity and interpretability.

First, we note that, $M_i(x)$ in (12) is an unnormalised triangular distribution and, therefore, nonnegative. We can integrate it in $[t_{i-1}, t_i]$ to get a monotone-increasing function

(an unnormalised CDF) $\mathcal{I}_i(x) = \int_{t_{i-1}}^{x} M_i(z) dx$:

$$\mathcal{I}_i(x) = \begin{cases} \frac{(x-t_{i-1})^2}{2(t_i - t_{i-1})} & t_{i-1} \le x \le t_i, \\ \frac{t_{i+1}-t_{i-1}}{2} - \frac{(t_{i+1}-x)^2}{2(t_{i+1}-t_i)} & t_i \le x \le t_{i+1}. \end{cases} \quad (16)$$

These are so-called I-splines [47]. Note the quadratic polynomials which play a similar role to the quadratic transformation in (15). The difference is that we do not transform the coefficients $\beta_i$, i.e., we still consider $f(x) = \sum_{i=1}^{l} \beta_i \mathcal{I}_i(x)$ thus preserving linearity. Therefore, we define the kernel

$$k(x,x') = \sum_{i=1}^{l} \sum_{j=1}^{l} \check{k}(t_i, t_j) \mathcal{I}_i(x) \mathcal{I}_j(x'). \quad (17)$$

**Theorem 3.** *Consider $l$ operational points $[u_1, \ldots, u_l]$ defined as $u_i = t_i$, then the SkewGP obtained from Theorem 1 with $L = diag([I_r, \mathbf{0}_r])$ and kernel defined as in (17) is monotone increasing in $[t_1, t_l]$.*

It is worth noticing that in this case we are imposing the monotonicity constraint through $[f(\mathbf{u}_1), \ldots, f(\mathbf{u}_r)] > 0$ which does not involve the derivatives. This is due to the choice of the I-spline basis function. This approach can also be applied to the multivariate case by using the same techniques discussed at the end of the previous section.

# 5 DESIRABILITY AS CLASSIFICATION

In desirability theory [57, 46, 5], decision making under uncertainty can be viewed as a choice between gambles. Formally, a gamble is a real-valued function on the possibility space: it represents a positive or negative pay-off that is uncertain in the sense that it depends on the unknown outcome. For instance, consider a simple coin toss, where the possible outcomes are Heads (H) and Tails (T). We can represent a gamble, $g$, as a two-dimensional vector, i.e. $g = [1, -2]$. This means you win 1 unit if it lands on Heads and lose 2 units if it lands on Tails. By choosing to accept or reject such gambles, a subject reveals their beliefs about the outcomes of the uncertain event. Consider buying a call-put option in finance as a tangible example of accepting a gamble.

Assume a subject has accepted the gambles $\mathscr{A} = \{g_1 = [1,0], g_2 = [0,1]\}$ and rejected $\mathscr{R} = \{g_3 = [-1,2], g_4 = [2,-1], g_5 = [-0.5, 3.5], g_6 = [3.5, -0.5]\}$, are they willing to accept the gamble $g_7 = [-1, 0]$?

This prediction task can be cast as a classification problem where we aim to predict the subject's acceptance (class 1) or rejection (class 0) of the gamble $g_7$. Here, consistency (rationality of the gambler) means that if the subject accepts the gamble $g = [g_a, g_b]$ they should also accept any gamble $g + h$ where $h > 0$ element-wise. Similarly, a subject should always reject gambles $g \leq 0$, because they are not favourable. These additional consistency constraints can be satisfied by finding a monotonic classifier that separates the augmented sets $\mathscr{A}' = \mathscr{A} \cup \{[-\varepsilon, -\varepsilon]\}$ and $\mathscr{R}' = \mathscr{R} \cup \{[0,0]\}$ for some small $\varepsilon > 0$. In linear desirability theory, we consider linear classifiers. It is well-known that linearity [62, 36, 16] is a strong assumption, being violated for instance in domains with budget constraints, problems with lack of liquidity, wealth effects and risk-aversion [38, 43, 44, 59]. We can then consider a more general nonlinear classifier and learn the subject's behaviour. Figure 6 shows in blue the region classified as 1 (accepted) for two nonlinear classifiers (we used the MSP kernel). It can be noted that the left one violates consistency: it implies the subject would accept negative gambles (third orthant) and reject positive gambles (first orthant). The right figure shows the accepted region after imposing monotonicity, which now satisfies consistency.[8] Note that, while this example employs the MSP kernel with an additive combination across the dimensions of the gamble, a product kernel could be used to capture interactions between the two dimensions.

---

[8]Technically, consistency holds within an error margin of $\varepsilon$. SkewGPs are continuous model. In the example, this means that around the origin $(0,0)$ the classifier may exclude some positive gambles, like $(\varepsilon/2, \varepsilon/2)$ for instance.

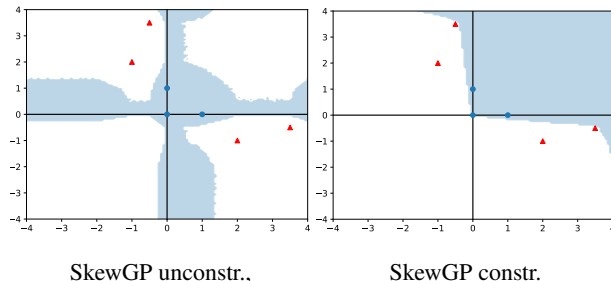

SkewGP unconstr.,          SkewGP constr.

Figure 6: Blue region: set of desirable gambles implied by an unconstrained classifier (left) versus a constrained classifier (right). The blue circles are the gambles in $\mathscr{A}$ and the red triangles those in $\mathscr{R}$.

Table 1: 1-D monotonic benchmark functions.

| | |
|---|---|
| $g_1(x) = 0.32(10x + \sin(10x))$ | $g_2(x) = 3(x < 0.8) + 6(x \geq 0.8)$ |
| $g_3(x) = 3x$ | $g_4(x) = 0.15e^{6x-3}$ |
| $g_5(x) = 3/(1 + e^{-20x+10})$ | $g_6(x) = 5x^2$ |
| $g_7(x) = 10(x+1)$ | |

# 6 NUMERICAL RESULTS

We assess the performance of our SkewGP formulation of monotonic constrained GP in simulated and real datasets. We will use the M-spline kernel defined in Section 4.2. Hyperparameters estimation is discussed in Appendix C. Appendix D provides an algorithmic description of the predictive posterior computation.

## 6.1 1D MONOTONIC BENCHMARK FUNCTIONS

To assess the performance of our SkewGP formulation for both regression and preference learning tasks, we leverage 7 established benchmark functions from prior works [27, 31, 51, 55], reported in table 1. As our method includes several previous approaches for imposing monotonicity (details in Section 4), we only compare it with monotonic-GP-flow (MF, [55]), which uses the numerical solution of a particular stochastic differential equation to impose monotonicity. Notably, we extend their model to preference learning by employing a probit likelihood.

For the regression task, the training data is generated by evaluating these functions at 100 randomly generated points in $[0,1]$ and adding independent Gaussian noise with variance $\sigma^2$ calculated using SNR=$\{10, 30\}$, that is: $\sigma^2 = $ signal variance/$SNR$. We generated 400 testing data from each $g_i$ to evaluate the performance of the models using the root-mean-square-error (RMSE) and the continuous-ranked-probability-score (CRPS) for the evaluation of probabilistic predictions. Table 2 shows the performance on test data evaluated with CRPS (lower is better) in the case SNR=10. SkewGPc denotes a monotonically

constrained model while SkewGPu is the unconstrained one.

Table 2: Results on regression task (CRPS).

| fun | MF | SkewGPu | SkewGPc |
|-----|-----|---------|---------|
| $g_1$ | 0.36±0.14 | 0.25± 0.03 | **0.15± 0.01** |
| $g_2$ | 0.82±0.12 | 0.74± 0.14 | **0.54±0.03** |
| $g_3$ | 0.28±0.17 | 0.19± 0.02 | **0.1±0.01** |
| $g_4$ | 0.44±0.11 | 0.19± 0.02 | **0.20±0.01** |
| $g_5$ | 0.62±0.16 | 0.29± 0.03 | **0.30±0.02** |
| $g_6$ | 0.56±0.31 | 0.31± 0.04 | **0.21±0.02** |
| $g_7$ | 1.80±0.44 | 1.96± 0.37 | **0.74± 0.21** |

For the preference task, the training data is generated by evaluating these functions at 50 randomly generated points in $[0,1]$ and then generating preference as $x_i \succ x_j$ if $g_l(x_i) + \varepsilon_i > g_l(x_j) + \varepsilon_j$ where $\varepsilon_i, \varepsilon_j$ are independent Gaussian noises with the same variance of the regression task. We generated 100 pairwise comparison between randomly selected $x_i$ in the training data. We also generated additional 400 pairwise comparison for testing and used the logarithmic-score (LogP) for the evaluation of probabilistic predictions. The definition of CRPS and LogP are provided in Appendix E.1 together with additional details about the numerical experiments. Table 3 shows the performances evaluated with LogP (higher is better) for the preference learning task with SNR=10. SkewGPc denotes a monotonically constrained model, SkewGPu the unconstrained one.

Table 3: Results on preference task (LogP).

| fun | MF | SkewGPu | SkewGPc |
|-----|-----|---------|---------|
| $g_1$ | -0.50± 0.02 | -1.04± 0.50 | -0.48± 0.09 |
| $g_2$ | -0.63± 0.04 | -0.88± 0.11 | -0.62± 0.05 |
| $g_3$ | -0.50± 0.03 | -1.04± 0.50 | **-0.45± 0.03** |
| $g_4$ | -0.52± 0.02 | -1.13± 0.06 | **-0.44± 0.04** |
| $g_5$ | -0.50± 0.02 | -0.92± 0.52 | **-0.40± 0.03** |
| $g_6$ | -0.47± 0.03 | -0.84± 0.37 | **-0.38± 0.08** |
| $g_7$ | -0.62± 0.02 | -0.96± 0.20 | -0.63± 0.03 |

In both regression and preference learning, it can be noticed that SkewGPc outperforms MF in probabilistic predictions. This is not fully surprising, because of the conjugacy of SkewGPs with both the normal and probit-affine likelihood. In Appendix E.1, we reported the timings for the algorithms and the results for SNR=30 and the RMSE.

## 6.2 SWISS ROUTE CHOICE DATA

In stated preference surveys, participants choose between options with trade-offs (like cost, time, or reliability), revealing their preferences in hypothetical scenarios. This approach is widely used in transportation for understanding how people value different features. We consider a dataset that includes the choices made by subjects regarding their preferred railway connections/routes in Switzerland. Each

scenario includes two alternatives described in terms of *travel time* (tt), *cost* (tc), *headway* (hw) and *number of interchanges* (ch) [56]. There are also subject specific variables: *household income*, *car-availability* (binary) and *purpose of the trip* (commute, shopping, business, leisure). Table 9 in Appendix shows a subset of the dataset. An example of a scenario where the subjects were asked to state their preference is:

$$Option1: \quad tt = 14, \ tc = 3, \ hw = 15, \ ch = 0,$$
$$Option2: \quad tt = 15, \ tc = 4, \ hw = 15, \ ch = 0.$$

It is clear that Option1 should be preferable to Option2. The dataset includes 3,492 pairwise preferences expressed by 388 individuals. In this type of analysis, it is common to learn a preference model for each group. For instance, hereafter we focus on commuters with car availability and compare an unconstrained SkewGP versus a constrained SkewGP, where we impose monotonicity (less is better) on all the covariates. We used 10-fold CV to compare the two models and we assessed the LogP score.

Table 4: Swiss route choice data (LogP).

| LogP | SkewGPu | SkewGPc |
|------|---------|---------|
| other-options | −0.53 | −0.53 |
| monotone-options | −0.30 | −0.15 |

Focusing on the options where one option is monotonically better than the other, SkewGPu achieves a worse average LogP value of −0.30 compared to −0.15 for SkewGPc, as shown in table 4. This is due to the uncertainty as shown in Figure 7 for the two options above. When one option is monotonically better than the other, SkewGPu often exhibits high uncertainty, predicting a utility difference near zero. In contrast, SkewGPc predicts the correct preference with high probability (the utility of the monotonically better option is always higher). As expected, the two models perform similarly for pairwise comparisons that are not monotonically dominated (LogP around −0.53).

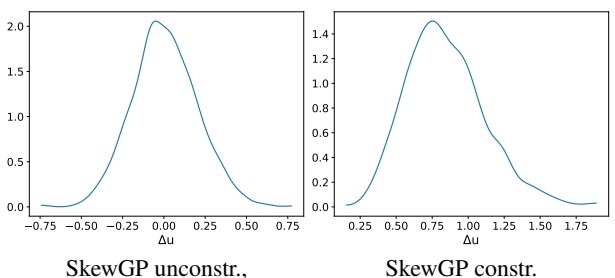

SkewGP unconstr.,                    SkewGP constr.

Figure 7: Posterior distribution of the utility difference between the two options. A positive difference denotes the correct prediction.

## 6.3 RISKY CHOICE DATASET

Understanding and predicting human decision-making becomes increasingly crucial as automated systems interact more closely with people. Building on this need, [45] collected a large dataset (10,000 preference-pairs) of human decisions. Each problem involved choosing between two gambles with distinct payoff-probability combinations.

$$Option1: \quad g_1 = [26, -1], \ p_1 = [0.95, 0.05],$$
$$Option2: \quad g_2 = [21, 23], \ p_2 = [0.95, 0.05].$$

For this choice-problem, 10 out of 15 subjects (67%) chose Option2. Expected Utility (EU) theory dictates that Option1 is preferable to Option2 because $26 \cdot 0.95 - 1 \cdot 0.05 = 24.65$ is higher than $21 \cdot 0.95 + 23 \cdot 0.05 = 21.1$. However, in Option2, we never lose money, so it is preferable in the worst case. There are other aspects to consider such as the way the payouts $g_{ij}$ are viewed by the decision-makers, if they use linearity when combining payoffs and probabilities, if they evaluate each gamble separately or jointly. However, also in this case, assuming monotonicity on $g$ seems to be reasonable: for instance, Option3 $g = [27, -0.5], \ p = [0.95, 0.05]$ should be preferable to Option1. Note that, this choice problem is related to desirability discussed in Section 5 - in desirability the probabilities are not given explicitly. We will use the dataset to learn a model to predict preferences for options by using $g_1, p_1, g_2, p_2, g_1 \cdot p_1, g_2 \cdot p_2$ as covariates. This will allow us to understand in which way the human choices deviates from EU theory. We will compare SkewGPu versus SkewGPc to understand the effect of monotonicity. We used 10-fold CV assessed the LogP score, the results are shown in table 5. We can reach similar conclusions to the ones for the Swiss route data: SkewGPc provides better estimates of the probability of the preference for monotone options.

Table 5: Risky choice data (LogP).

| LogP | SkewGPu | SkewGPc |
|---|---|---|
| other-options | $-0.31$ | $-0.31$ |
| monotone-options | $-0.17$ | $-0.13$ |

Table 6 shows that SkewGPc outperforms the EU model in terms of accuracy. This suggests that the underlying preferences in the dataset deviate from the linear assumptions of the EU model, and a nonlinear model like SkewGPc is more appropriate for capturing these preferences.

Table 6: Risky choice data, comparison with EU (LogP).

| Accuracy | EU | SkewGPc |
|---|---|---|
| other-options | 0.75 | 0.83 |
| monotone-options | 0.97 | 0.99 |

## 7 CONCLUSIONS

We derived a unified framework for linearly constrained Gaussian Processes (GPs) by using Skew GPs which includes regression, classification, and preference learning. Our unified framework demonstrated strong performance in both preference learning and modelling human decision-making. As future work, we aim to apply this approach to active learning and Bayesian optimisation, while including a larger class of linear constraints, beyond monotonicity. For human-decision making under risk, we plan to derive application-specific basis functions and kernels and impose constraints that are usually assumed in decision making, such as both monotonicity and convexity.

### Acknowledgements

For the first author, this publication has emanated from research conducted with the financial support of the EU Commission Recovery and Resilience Facility under the Science Foundation Ireland Future Digital Challenge Grant Number 22/NCF/FD/10827. The second author acknowledges support from the SNSF grant number 200021_212164.

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

# Linearly Constrained Gaussian Processes are SkewGPs: application to Monotonic Preference Learning and Desirability
# (Supplementary Material)

**Alessio Benavoli**[1]                    **Dario Azzimonti**[2]

[1]School of Computer Science and Statistics, Trinity College Dublin, Ireland
[2]Dalle Molle Institute for Artificial Intelligence (IDSIA), USI/SUPSI, Lugano, Switzerland

## A   GAUSSIAN PROCESSES

Gaussian Processes (GPs) are prior over functions [40, 48], that have attractive advantages over parametric (including neural networks) models.[1] They have a small number of tunable hyperparameters (and so they can be trained on small datasets), and give a measure of prediction uncertainty. Moreover, by being kernel based, they provide a framework to learn utility functions defined on any domain $\mathscr{X}$ on which we can define a kernel-function.

To define a prior over a function $f : \mathscr{X} \to \mathbb{R}$, a GP assumes that, for every $n$, $p(f(\mathbf{x}_1), \ldots, f(\mathbf{x}_n))$ is jointly Gaussian, with mean $[\mu(\mathbf{x}_1), \ldots, \mu(\mathbf{x}_n)]$ and covariance $Cov(f(\mathbf{x}_i), f(\mathbf{x}_j)) = k(\mathbf{x}_i, \mathbf{x}_j)$, for $i, j = 1, \ldots, n$. $\mu(\mathbf{x})$ and $k(\mathbf{x}, \mathbf{x}')$ are the mean function and, respectively, the (positive definite) kernel function of the GP. A GP is usually parameterised with a zero mean function $\mu(\mathbf{x}) = 0$ and a covariance kernel $k_{\boldsymbol{\theta}}(\mathbf{x}, \mathbf{x}')$ which depends on hyperparameters $\boldsymbol{\theta} \in \Theta$. A typical example is the automatic relevance determination (ARD) square-exponential kernel on $\mathbb{R}^c$, $c \in \mathbb{N}$. For $\mathbf{x}, \mathbf{x}' \in \mathbb{R}^c$ it is defined as

$$k_{\boldsymbol{\theta}}(\mathbf{x}, \mathbf{x}') = \sigma_0^2 \exp\left( -\sum_{i=1}^{c} \frac{(x_i - x_i')^2}{2\ell_i^2} \right), \tag{18}$$

where $\boldsymbol{\theta} = [\ell_1, \ldots, \ell_c, \sigma_0^2]$ includes the lengthscales hyperparameters $\ell_i$ (one for each dimension) and the scale parameter $\sigma_0^2$. GPs have a natural Bayesian interpretation that makes them ideal for regression problems. If we assume that the observed values are the sum of a true function evaluated at some inputs plus Gaussian noise, i.e. $y_i = f(\mathbf{x}_i) + \varepsilon_i$ with $\varepsilon_i \sim N(0, \sigma^2)$ for $i = 1, \ldots, n$, then we can analytically compute the posterior distribution of $f$. We can write the observation model more compactly as the likelihood

$$p(y_1, \ldots, y_n | f(\mathbf{x}_1), \ldots, f(\mathbf{x}_n)) = N(\mathbf{y}_n | \mathbf{f}(X), \sigma^2 I_n),$$

where $\mathbf{y}_n = [y_1, \ldots, y_n]^\top$, $X = [\mathbf{x}_1, \ldots, \mathbf{x}_n]^\top$ and $I_n$ is the identity matrix of dimension $n$. In particular, the predictive posterior at a new test point $\mathbf{x}^* \in \mathscr{X}$ is $GP(\mu_p, k_p)$, with mean and covariance kernel given by:

$$\mu_p(\mathbf{x}^*) = K_{\boldsymbol{\theta}}(\mathbf{x}^*, X)(K_{\boldsymbol{\theta}}(X, X) + \sigma^2 I_n)^{-1} \mathbf{y}_n \tag{19}$$

$$k_p(\mathbf{x}^*, \mathbf{x}^*) = K_{\boldsymbol{\theta}}(\mathbf{x}^*, \mathbf{x}^*) - K_{\boldsymbol{\theta}}(\mathbf{x}^*, X)(K_{\boldsymbol{\theta}}(X, X) + \sigma^2 I_n)^{-1} K_{\boldsymbol{\theta}}(X, \mathbf{x}^*), \tag{20}$$

where $K_{\boldsymbol{\theta}}(X, X)$ is a matrix whose ij-th element is defined as $(K_{\boldsymbol{\theta}}(X, X))_{ij} = k_{\boldsymbol{\theta}}(\mathbf{x}_i, \mathbf{x}_j)$ (similar for $K_{\boldsymbol{\theta}}(\mathbf{x}^*, X)$). Note that, the variance of the likelihood $\sigma^2$ is also considered to be a hyperparameter. The hyperparameters $\boldsymbol{\theta}, \sigma^2$ are commonly estimated by maximising the marginal likelihood:

$$p((\mathbf{x}_i, y_i) | \boldsymbol{\theta}, \sigma^2) = N(\mathbf{y}_n, K_{\boldsymbol{\theta}}(X, X) + \sigma^2 I_n).$$

In tasks with likelihoods different from the Gaussian, the posterior is not a GP. For Probit (classification/preference learning) and Skew-Normal likelihoods, the posterior is a Skew GP [8]. For other likelihoods, in general the posterior does not

---

[1]GPs can be seen as single-layer neural networks with an infinite number of hidden units [60].

have a closed-form and is approximated with a GP using three main approaches: (i) Laplace Approximation (LP) [32, 61]; (ii) Expectation Propagation (EP) [35]; (iii) Kullback-Leibler divergence (KL) minimization [41], comprising Variational Bounding (VB) [21] as a particular case.

## B  PROOFS

The proofs are straightforward.

**Theorem 1**  By denoting with $\mathbf{z}_1 = f(\mathbf{x})$ and $\mathbf{z}_0 = [\mathbf{f}(\mathbf{u}_1), \ldots, \mathbf{f}(\mathbf{u}_r), \mathbf{f}'(\mathbf{u}_1), \ldots, \mathbf{f}'(\mathbf{u}_r)]^\top$, we can rewrite the constraint (9) as $L\mathbf{z}_0 + \boldsymbol{\gamma} > 0$. Therefore, the distribution of $\mathbf{z}_1 + \boldsymbol{\xi}(\mathbf{x}) = f(\mathbf{x}) + \boldsymbol{\xi}(\mathbf{x})$ conditioned on $L\mathbf{z}_0 + \boldsymbol{\gamma} > 0$ is SkewGP as derived in Section 2. We just need to consider a change of variables to take into account of the matrix $L$.

**Theorem 2**  Consider $f(x) = \sum_{i=1}^{l} \beta_i M_i(x)$ and $x' \in (t_{i-1}, t_i)$ and observe that $f(x') = \beta_{i-1}(t_i - x')/(t_i - t_{i-1}) + \beta_i(x' - t_{i-1})/(t_i - t_{i-1})$. Therefore, we have that

$$\frac{d}{dx} f(u_i) = (\beta_i - \beta_{i-1}) \frac{t_i}{t_i - t_{i-1}}.$$

Therefore, we have that $\frac{d}{dx} f(u_i) > 0$ implies that $\beta_i - \beta_{i-1} > 0$ for $i = 1, \ldots, l$, which is equivalent to the constraint [28, Eq. (7)].

**Theorem 3**  Consider $f(x) = \sum_{i=1}^{l} \beta_i \mathscr{I}_i(x)$ and $x' \in (t_i, t_{i+1})$ and observe that

$$f(x') = \beta_i \left( \frac{t_{i+2} - t_i}{2} - \frac{(t_{i+1} - x')^2}{2(t_{i+1} - t_i)} \right)$$
$$+ \beta_{i+1} \frac{(x' - t_i)^2}{2(t_{i+1} - t_i)}$$

Therefore, we have that $f(u_i) = \beta_{i-1} \left( \frac{t_{i+2} - t_i}{2} - \frac{t_{i+1} - t_i}{2} \right)$. Therefore, we have that $f(u_i) > 0$ implies that $\beta_i > 0$ for $i = 1, \ldots, l$ which ensures monotonicity.

## C  HYPERPARAMETERS' ESTIMATION

We use the implementation of Variational Inference in *GPytorch* [19] to estimate the kernel hyperparameters. This is based on [24] although in our case the inducing points are equal to the set of the covariates $X$ plus the operational points (we perform a full variational inference). We apply the variational inference considering as prior the Multivariate Normal in (1) and we include the constraint $L\tilde{\mathbf{f}} + \boldsymbol{\gamma} > 0$ with

$$\tilde{\mathbf{f}} = \begin{bmatrix} \mathbf{f}(\mathbf{u}_1) \\ \vdots \\ \mathbf{f}(\mathbf{u}_r) \\ \mathbf{f}'(\mathbf{u}_1) \\ \vdots \\ \mathbf{f}'(\mathbf{u}_r) \end{bmatrix}, \tag{21}$$

into the likelihood through a probit $\Phi(\frac{1}{\tau}(L\tilde{\mathbf{f}} + \boldsymbol{\gamma}))$ so to make the gradient to be continuous. $\tau$ is a constant. It is well known that for $\tau \to 0$ the Gaussian CDF converges to an indicator function for its argument being positive, that is $L\tilde{\mathbf{f}} + \boldsymbol{\gamma} > 0$. Therefore, we choose $\tau = 10^{-3}$ and we decrease it during the maximisation of ELBO in order to get even closer to the indicator function (from $\tau = 10^{-3}$ up to $\tau = 10^{-6}$). Note that, we use this approximation of the constraint only for estimating the kernel hyperpameters. The samples from the posterior are computed through the SkewGP derivations in Proposition 1 and Theorem 1.

We fix the operational points for the SE kernel and knots for the MSP kernel to $n$ percentiles of the data and we do not change them during hyperparameter optimisation. Approaches to optimally placing the operational points has been discussed in previous literature [49, 58, 1, 15, 22, 30, 29, 28, 31].

# D  SAMPLE FROM THE CONSTRAINED PREDICTIVE POSTERIOR

Algorithm 1 details how samples from the predictive posterior of SkewGPc are obtained. Note that the posterior parameters (lines 1 and 2) and the truncated normal sampling (line 3) are computed once for all as they do not depend on $\mathbf{x}^*$. The steps at line 4-5 are sampling from a multivariate Gaussian and matrix-vector multiplications which are fast operations.

---

**Algorithm 1:** Predictive posterior for SkewGPc

---

**Data:** $k$ kernel function, $U \in \mathbb{R}^{r \times D}$ matrix of operational points, $L$ matrix specifying monotonicity constraints, $W \in \mathbb{R}^{m_a \times n}$ matrix of preference data, $m$ number of posterior samples, $\mathbf{x}^*$ new input.

1  Compute the prior constrained parameters $\Gamma$, $\Delta(\mathbf{x})$ by using eqs. (10) and (11);

2  Compute the posterior functions and parameters $\tilde{\boldsymbol{\xi}}(\mathbf{x})$, $\tilde{\Omega}(\mathbf{x},\mathbf{x}')$, $\tilde{\Delta}(\mathbf{x})$, $\tilde{\boldsymbol{\gamma}}$, $\tilde{\Gamma}$ as in Prop. 1 ;

3  Sample $\mathbf{r}^*_{1,-\tilde{\gamma}}$ from the multivariate normal $N(0,\tilde{\Gamma})$ truncated below $\tilde{\boldsymbol{\gamma}}$ by using *lin-ess* ;

4  At the predictive input point $\mathbf{x}^*$, sample $\mathbf{r}^*_0$ from the multivariate normal $N(0,\overline{\tilde{\Omega}}(\mathbf{x}^*,\mathbf{x}^*) - \tilde{\Delta}(\mathbf{x}^*)\tilde{\Gamma}^{-1}\tilde{\Delta}(\mathbf{x}^*)^T)$ ;

5  Compute $\tilde{\mathbf{z}} = \tilde{\boldsymbol{\xi}}(\mathbf{x}^*) + \mathbf{r}^*_0(\mathbf{x}^*) + \Delta(\mathbf{x}^*)\Gamma^{-1}\mathbf{r}^*_{1,-\tilde{\gamma}}$ ;

6  **return** *Samples* $\tilde{\mathbf{z}}$;

---

# E  NUMERICAL RESULTS

## E.1  1D SIMULATIONS

The logarithmic score is used to evaluate probabilistic prediction for binary observations. Consider a variable $y$ with possible values 1 or 0, denote the probability of $y=1$ with $p$, then one can write the logarithmic scoring rule as $LogP(p) = y\ln(p) + (1-y)\ln(1-p)$. Since we are comparing Bayesian methods, we computed the average logarithmic score by averaging over the $S$ samples from the posterior

$$LogP(y,p_1,\ldots,p_S) = \sum_{i=1}^{S} y\ln(p_i) + (1-y)\ln(1-p_i)$$

The continuous ranked probability score (CRPS) is a strictly proper scoring rule much used to assess probabilistic prediciton for continuous variables. It is defined as

$$CRPS(F,y) = \int_{\mathbb{R}} (F(x) - H(x \geq y))^2 dx$$

where $F$ is the predicted cumulative distribution function, $H$ is the Heaviside step function and $y \in \mathbb{R}$ is the observation. We computed the CRPS using the empirical CDF computed from the posterior samples and we used as $y$ the true value of the function (and not the noisy one).

For both monotonic-flow and SkewGP we used 20 inducing points and, respectively, Knots. We initialised them with the percentiles of the data. For monotonic-flow, we used the SE kernel and T=1. For both the models, in regression, we standardised the $y$s before 'training'.

Table 7 reports the results with SNR=30 for the regression task (CRPS, lower better) and the preference task (LogP, higher better).

In regression, it can be noted that SkewGPu and SkewGPc have a similar performance (the performance of the latter has higher variability). This is mainly due to the small SNR (SkewGPu can learn the monotonicity of the functions from the data) and variability of the numerical optimisation.

For preference learning, to compute the kernel hyperparameters and predict 2000 posterior samples, the running (wall-clock) time is 330s for MF and 166s for SkewGPc.

For fixed hyperparameters, to predict 2000 posterior samples, the running (wall-clock) time is 180s for MF and 5s for SkewGPc. Sampling with Monotonic GPflow requires solving a stochastic differential equation numerically.

Finally, table 8 reports the RMSE $\sqrt{\frac{1}{n}\sum_{i=1}^{n}(g_j(x_i) - \hat{g}_j(x_i))^2}$ where $\hat{g}_j$ is the posterior mean, for $j = 1,\ldots,7$, for both the SNR 10 and 30 case.

Table 7: 1-D benchmark functions results with SNR=30.

| | CRPS regression | | | LogP preference | | |
|---|---|---|---|---|---|---|
| fun | MF | SkewGPu | SkewGPc | MF | SkewGPu | SkewGPc |
| $g_1$ | 0.59±0.04 | 0.19± 0.01 | **0.21± 0.06** | -0.46± 0.04 | -0.82± 0.40 | **-0.34± 0.03** |
| $g_2$ | 0.89±0.08 | 0.58± 0.10 | **0.53±0.06** | -0.62± 0.04 | -0.88± 0.21 | -0.58± 0.08 |
| $g_3$ | 0.50±0.05 | 0.14± 0.01 | **0.16±0.05** | -0.44± 0.04 | -0.91± 0.40 | **-0.33± 0.05** |
| $g_4$ | 0.50±0.05 | 0.18± 0.01 | **0.20±0.03** | -0.49± 0.03 | -0.95± 0.46 | **-0.40± 0.06** |
| $g_5$ | 0.79±0.06 | 0.25± 0.02 | **0.34±0.11** | -0.49± 0.03 | -0.83± 0.34 | **-0.36± 0.04** |
| $g_6$ | 0.99±0.07 | 0.25± 0.02 | **0.34±0.14** | -0.43± 0.04 | -0.94± 0.48 | **-0.32± 0.08** |
| $g_7$ | 1.70±0.33 | 0.95± 0.2 | **0.57± 0.2** | -0.55± 0.02 | -0.88± 0.20 | -0.53± 0.04 |

Table 8: Regression benchmark functions RMSE results.

| | RMSE SNR 10 | | | RMSE SNR 30 | | |
|---|---|---|---|---|---|---|
| fun | MF | SkewGPu | SkewGPc | MF | SkewGPu | SkewGPc |
| $g_1$ | 0.55±0.17 | 0.43±0.05 | **0.25±0.02** | 0.78±0.06 | **0.33±0.026** | 0.35±0.1 |
| $g_2$ | 1.1±0.1 | 1.14±0.11 | **0.88±0.05** | 1.11±0.17 | 1.0±0.14 | **0.86±0.04** |
| $g_3$ | 0.44±0.24 | 0.35±0.04 | **0.11±0.05** | 0.71±0.05 | **0.25±0.03** | **0.25±0.1** |
| $g_4$ | 0.6±0.1 | 0.41±0.05 | **0.35±0.02** | 0.64±0.04 | **0.38±0.04** | **0.38±0.06** |
| $g_5$ | 0.92±0.21 | **0.49±0.06** | 0.53±0.05 | 1.1±0.09 | **0.41±0.03** | 0.67±0.12 |
| $g_6$ | 0.82±0.37 | 0.57±0.09 | **0.35±0.05** | 1.28±0.09 | **0.46±0.06** | 0.55±0.22 |
| $g_7$ | 2.67±0.54 | 2.7±0.3 | **1.30±0.46** | 2.5±0.4 | 1.65±0.3 | **0.90±0.48** |

SkewGPu outperforms MF. Note that, the RMSE for SNR=10 is sometimes lower than for SNR=30. This counter intuitive result occurs because both RMSE and CRPS were calculated on the noise-free function values.

Our simulation results align with [55], where the method [3] achieved similar or even better performance than MF in some benchmark function. As previously discussed, MSP kernels based on M-splines or MSP kernels based on I-splines provide an improvement over [3] by preserving conjugacy with both the normal and affine-probit likelihood.

## E.2 SWISS ROUTE CHOICE DATA

In the MSP kernel, we used 10 knots per covariate.

| ID | choice | tt1 | tc1 | hw1 | ch1 | tt2 | tc2 | hw2 | ch2 | hh_inc_abs | car_availability | commute | shopping | business | leisure |
|---|---|---|---|---|---|---|---|---|---|---|---|---|---|---|---|
| 2439 | 2 | 58 | 7 | 30 | 1 | 50 | 8 | 30 | 0 | 50000 | 1 | 1 | 0 | 0 | 0 |
| 2439 | 1 | 30 | 8 | 60 | 0 | 41 | 7 | 15 | 2 | 50000 | 1 | 1 | 0 | 0 | 0 |
| 2439 | 1 | 41 | 7 | 30 | 0 | 34 | 8 | 15 | 2 | 50000 | 1 | 1 | 0 | 0 | 0 |
| 2439 | 1 | 44 | 10 | 60 | 1 | 52 | 9 | 60 | 2 | 50000 | 1 | 1 | 0 | 0 | 0 |
| 2439 | 2 | 43 | 9 | 60 | 0 | 34 | 10 | 30 | 0 | 50000 | 1 | 1 | 0 | 0 | 0 |
| 2439 | 2 | 36 | 8 | 60 | 1 | 43 | 7 | 15 | 1 | 50000 | 1 | 1 | 0 | 0 | 0 |
| 2439 | 2 | 30 | 8 | 60 | 0 | 43 | 7 | 15 | 0 | 50000 | 1 | 1 | 0 | 0 | 0 |
| 2439 | 1 | 43 | 8 | 30 | 1 | 30 | 9 | 60 | 0 | 50000 | 1 | 1 | 0 | 0 | 0 |
| 2439 | 1 | 41 | 8 | 30 | 2 | 58 | 7 | 60 | 0 | 50000 | 1 | 1 | 0 | 0 | 0 |
| 5641 | 1 | 77 | 19 | 15 | 1 | 110 | 16 | 60 | 1 | 10000 | 0 | 0 | 0 | 0 | 1 |
| 5641 | 2 | 94 | 23 | 60 | 1 | 125 | 18 | 15 | 0 | 10000 | 0 | 0 | 0 | 0 | 1 |
| 5641 | 2 | 82 | 18 | 60 | 2 | 91 | 15 | 30 | 0 | 10000 | 0 | 0 | 0 | 0 | 1 |
| 5641 | 2 | 101 | 15 | 60 | 0 | 86 | 20 | 15 | 0 | 10000 | 0 | 0 | 0 | 0 | 1 |
| 5641 | 1 | 99 | 18 | 15 | 0 | 110 | 16 | 15 | 0 | 10000 | 0 | 0 | 0 | 0 | 1 |
| 5641 | 1 | 91 | 18 | 30 | 1 | 101 | 16 | 15 | 0 | 10000 | 0 | 0 | 0 | 0 | 1 |

Table 9: 10 pairwise options: each rows is a different scenario: tt1,tc1,hw1,ch1 against tt2,tc2,hw2,ch2. Choice denotes the option selected by th user ID.

## E.3 RISKY CHOICE DATA

We only focused on choice-pairs whose gambles have only two components (two-dimensional), as in the example in Section 6.3: a total of 5347 choices.