# OpenReview forum: "Linearly Constrained Gaussian Processes are SkewGPs: application to Monotonic Preference Learning and Desirability"
_auai.org/UAI/2024/Conference — UAI 2024 poster_

### Official Review · Reviewer_knMT · 2024-03-20

**Q2-1 Originality-Novelty:** 3
**Q2-2 Correctness-Technical Quality:** 3
**Q2-5 Clarity Of Writing:** 4

**Q1 Summary And Contributions:**

The paper deals with Gaussian processes imposed by inequality constraints. By referring to prior art where these constraints have been imposed on a finite set of operational points, the paper contributes with an insight that this can be seen as Skew Gaussian processes. Exploiting the conjugacy of Skew GPs enables, to extend beyond regression, which has been the primary focus of previous work on linearly constrained GPs.

**Q2-3 Extent To Which Claims Are Supported By Evidence:**

3: Good: the main claims are supported by convincing evidence (in the form of adequate experimental evaluation, proofs, (pseudo-)code, references, assumptions).

**Q2-4 Reproducibility:**

2: Fair: key resources (e.g. proofs, code, data) are unavailable but key details (e.g. proof sketches, experimental setup) are sufficiently well-described for an expert to confidently reproduce the main results.

**Q3 Main Strengths:**

From what I can judge, the paper is technically correct and clear writing. Proofs are available, despite short, they seem to be self-contained. The paper is well-organized and clearly written.

**Q4 Main Weakness:**

The conjugacy of Skew Gaussian processes is promising, but details of how this is exploited could be more pronounced in connections to the experimental part of the paper. Code is not available, but I do think that an expert would be able to reproduce the main results based on the description. In some aspects it is not clear how the paper differs from prior art.

**Q5 Detailed Comments To The Authors:**

I think it would be informative for the reader to already in the abstract and possibly title be clearer that you consider inequality constraints and not linear (equality) constraints in general. You list the exploitation of conjugacy for SkewGPs as one of the contributions. I think the paper could explain better how this conjugacy is exploited in the experimental part of the paper.

In Section 4.2 impose constrains in the whole domain as was proposed in [30,29,28,31]. The paper would benefit from being clear on how it is different to these prior works. I suppose the main difference is that they don’t do the connection to Skew GPs, and hence don’t exploit the conjugacy, which connects back to my previous comment. In section 6.1 you only compare with monotonic-GP-flow since your method includes several previous approaches for imposing monotonicty. But don’t you have computational benefits over [30,29,28,31] that could be mentioned if these references don’t exploit conjugacy?

Minors comments:
* Page 2, two rows below eq (1): An N is missing in the conditional normal.
* Figure 4: Unclear how the numbers on x-axis relate to t_0, t_1,… Suggestion to use t_0,,t_1,.. as ticks on the x-axis instead of numbers.
* Theorem 2 and 3, third line: ) missing after diag-expression.
* Page 6: witk → with
* Sometimes you write k(x,x) for kernel, and sometimes k(x,x’)
* Section 6.1: You denote monotinic-GP-flow with the abbreviation MP, but in table and text afterwards, you abbreviate with MF.
* Appendix B, proofs: You have f(x’) on LHS, but x on RHS when describing how f depends on its two neighboring basis functions.

**Q9 Complying With Reviewing Instructions:**

Yes

---

> ### Author Rebuttal · Authors · 2024-04-04
>
> We thank the reviewer for their detailed comments. We will make sure to include all minor comments in the revised version of the paper.
>
> 1. **Inequality, not equality, constraints.** We will further clarify in the introduction that we consider inequality constraints and not linear equality constraints.
>
> 2. **Computational benefits of skewGP formulation.** As the reviewer correctly points out our model includes the models in [30,29,28,31] as starting from a GP prior and encoding the constraints inside the skewGP structure we derive an exact posterior for the model.
>
> The reviewer's observation is indeed valid; we do gain computational advantages by leveraging conjugacy. For instance, in reference [28], the authors determine the mode of the posterior to derive the mean of the posterior distribution at the training data. Subsequently, to predict the mean at the testing data and compute uncertainty, they employ a Hamiltonian Monte Carlo method (initialized at the mode).
>
> A similar approach is used in [30], but here the authors utilise a tailored rejection sampling algorithm for convex sets to sample from the predictive posterior distribution.
>
>
> In our approach, computing the mode of the posterior mean is unnecessary due to the availability of analytical derivations. This allows us to determine the parameters of the posterior SkewGP, which we then utilise for sampling. Consequently, we are relieved of the computational burden associated with computing the mode. Furthermore, tailored MCMC methods are unnecessary in our approach. We use the same MCMC method across all the examples and numerical experiments. Indeed, with conjugacy, we can straightforwardly compute the posterior for the approaches outlined in Sections 4.1 and 4.2 by simply changing the Kernel and the location of the operational points. In a nutshell, SkewGP allows the user to seamlessly incorporates linear inequality constraints into GPs in a transparent way, thanks to its analytical derivations. We firmly believe that this is an important contribution to linearly constrained GPs.
>
> Alternative methods, such as those discussed in reference [1], resort to approximating the posterior via Expectation Propagation. However, as noted (in the case without monotonicity constraints) by Takeno et al. (see reference below), EP tends to offer a bad approximation of the posterior in the context of preference learning compared to SkewGPs. Laplace's and Variational approximations perform even worse.
>
> Takeno, Shion, Masahiro Nomura, and Masayuki Karasuyama. "Towards practical preferential Bayesian optimization with skew Gaussian processes." International Conference on Machine Learning. PMLR, 2023.
>
> We will discuss  this aspect in the revised version. In order to clarify the importance of the analytical derivations, we will include in the Appendix the step-by-step algorithm we use to compute the predictive posterior samples for SkewGP with monotonicity constraints (which combines the results of Proposition 1 and Theorem 1). This will clarify that our approach is Kernel and constraints-agnostic and clarify the implementation of the method we use in the numerical experiments.

---

### Official Review · Reviewer_KCUW · 2024-03-20

**Q2-1 Originality-Novelty:** 3
**Q2-2 Correctness-Technical Quality:** 3
**Q2-5 Clarity Of Writing:** 4

**Q1 Summary And Contributions:**

The authors extend skewGPs to handle monotonicity constraints.

**Q2-3 Extent To Which Claims Are Supported By Evidence:**

3: Good: the main claims are supported by convincing evidence (in the form of adequate experimental evaluation, proofs, (pseudo-)code, references, assumptions).

**Q2-4 Reproducibility:**

3: Good: key resources (e.g. proofs, code, data) are available and key details (e.g. proofs, experimental setup) are sufficiently well-described for competent researchers to confidently reproduce the main results.

**Q3 Main Strengths:**

Well written with nice figures that supplement text throughout

**Q4 Main Weakness:**

Weaknesses stated as one/few liners or ignored (e.g., sum kernels, potential univariate requirement for Theorem 3)

MF evaluation missing with real data

**Q5 Detailed Comments To The Authors:**

pg 3: "This likelihood encompasses all the standard likelihood functions used in regression, classification and preference-learning." Would clarify that this holds in GPs only. The probit likelihood is for instance not the go to likelihood in linear binary classification - most use logistic regression.

pg 4 " We use a variational inference technique to approximate the posterior distribution with a Gaussian distribution. " Would reference Appendix C here.

Section 4.1: you call the kernels squared exponential, but then refer to them as RBF kernels in the Figure. Would keep notation consistent.

Section 4.2: why cant you just take a semi supervised approach by including the test set (without labels) as operational points to enforce monotonicity over the test set too?

Theorem 2: I was going to recommend including intuition, but then I noticed that the proofs were really short. I would point the reader to Appendix B and tell them that the proofs are short so they can gain intuition about why the conclusion holds. Right now, Theorem 2 is just stated as a sudden fact.

End of Section 4.2: why cant you just consider the product kernel? where does the theory break down in this case? please explain in the paper. usually multidimensional splines are handled by taking the product of the individual kernels rather than their sum.

Theorem 3: explain if this also works for the sum kernel like in Section 4.2. Looking at the proof, it only seems to work for univariate X, but I may be mistaken.

Figure 6: the sum kernel cannot handle XOR problems, but this should not be a problem under consistency right? If so, this would be important to explain to the reader.

Section 6.1 "monotonic-GP-flow (MP)" --> MF

Why didnt you include MF with the real data?

Section 4.1 "This translates to fast inference" Would recommend including timing results of all tested algorithms in the experiments to support this statement.

**Q9 Complying With Reviewing Instructions:**

Yes

---

> ### Author Rebuttal · Authors · 2024-04-04
>
> We would like to thank the reviewer for their detailed comments. We will make sure to include all the reviewer's suggestion in the revised version of the paper. Regarding the reviewer's question we answer below point by point:
>
> 1. **Semi-supervised approach:** it is an interesting approach that we have not yet considered. There are indeed applications where the test points are known in advance and one could impose monotonicity at the test points. This could be useful for the approach we discuss in Section 4.1. However, for the approach in Section 4.2, the constraints are already guaranteed to be satisfied at  all xs, no matter the location of the operational points.
>
> 2. **Additive vs product kernel:** We thank you the reviewer for the comment. It is  actually straightforward to include the product kernel in our approach similarly to what done by
>
> Hassan Maatouk and Xavier Bay. Gaussian process emulators for computer experiments with inequality constraints. Mathematical Geosciences, 49:557–582, 2017
>
> and as well as using an ANOVA kernel (including both sum and products). We will clarify this in the revised version. The additive kernel holds the advantage of scaling more effectively to high dimensions.
>
> 3. **Theorem 3:** this theorem does indeed require a univariate input, however the kernel in eq. 17 can be generalized to the D-dimensional case with an additive kernel. We will add this comment in the revised version of the paper.
>
> 4. **XOR problems and sum kernels:** The Reviewer is right, we will clarify this in the revised version.
>
> 5. **Include MF in the real-datasets:** We haven't included the monotonic flow kernel in the real-datasets examples due to its inefficiency in running across multiple dimensions. Additionally, we encountered challenges in setting the optimal values for its hyperparameters (the process was not converging). Note that, the original paper on Monotonic Gaussian Process Flows by Ustyuzhaninov et al. only explored one-dimensional or multi-dimensional (but independently across features) scenarios.
>
> 6. **Timings for the algorithms:** We will include the running time in the revised version. For preference learning, to compute the kernel hyperparameter and predict 2000 posterior samples, the running (wall-clock) time is:
>
> Monotonic GPflow: 330 seconds
>
> SkewGP: 166 seconds
>
> For fixed hyperparameters, to predict 2000 posterior samples, the running (wall-clock) time is:
>
> Monotonic GPflow: 180 seconds
>
> SkewGP: 5 seconds
>
> Sampling with Monotonic GPflow requires solving a stochastic  differential equation numerically.

---

### Official Review · Reviewer_7AW2 · 2024-03-21

**Q2-1 Originality-Novelty:** 2
**Q2-2 Correctness-Technical Quality:** 3
**Q2-5 Clarity Of Writing:** 1

**Q1 Summary And Contributions:**

This work focuses on the preference tasks from the perspective of GP. There are two main steps: 1. build the linearly constrained GP as SKewGPs; 2. borrow the unified theory from SkewGPs to analyze monotonic preference learning and desirability.

The concerned topic is interesting. However, I worry about, if a random process is used to induce preferences, will its expressive ability be too weak?

The theoretical results look like solid.

The novelty, especially technological innovation, of this work is limited.

The main drawback is the poor writing and organization of this paper, which heavily hamper the readability and understanding of this work.

Besides, I'm not sure what the purpose of this experiment is? What do authors want to illustrate through figures and tables? I don't quite understand the design logic of this verification experiment. In my opinion, the connection between this experiment and the theoretical conclusions of this paper is weak.

**Q2-3 Extent To Which Claims Are Supported By Evidence:**

2: Fair: the main claims are somewhat supported by evidence (but the experimental evaluation may be weak, or does not match entirely with the claims, important baselines may be missing, proofs contain important ideas but lack rigor, algorithmic details are only discussed superficially, references are imprecise, assumptions are not sufficiently motivated or explicated, etc.).

**Q2-4 Reproducibility:**

3: Good: key resources (e.g. proofs, code, data) are available and key details (e.g. proofs, experimental setup) are sufficiently well-described for competent researchers to confidently reproduce the main results.

**Q3 Main Strengths:**

As mentioned above.

The concerned topic is interesting.

The theoretical results look like solid.

The novelty, especially technological innovation, of this work is limited.

**Q4 Main Weakness:**

The main drawback is the poor writing and organization of this paper, which heavily hamper the readability and understanding of this work.

Besides, I'm not sure what the purpose of this experiment is? What do authors want to illustrate through figures and tables? I don't quite understand the design logic of this verification experiment. In my opinion, the connection between this experiment and the theoretical conclusions of this paper is weak.

**Q5 Detailed Comments To The Authors:**

I worry about, if a random process is used to induce preferences, will its expressive ability be too weak?

**Q9 Complying With Reviewing Instructions:**

Yes

---

> ### Author Rebuttal · Authors · 2024-04-04
>
> We thank the reviewer for the detailed review, we will answer point by point below.
>
> 1. **Novelty:** We would like to stress that, to the best of our knowledge, in the literature of Gaussian processes with monotonic constraints, the connection between operational points based methods and skewGP is new, which is a novel theoretical result. Indeed, this connection allows for an analytical calculation of the posterior distribution in both the regression and the preference case. This allows to naturally include monotonic constraints in preferential learning, which is also a novel theoretical result.
>
> 2. **Purpose of the experimental section.** The experimental section presents three experiments: section 6.1 shows that modelling monotonic constraints with a skewGP provides better performances than monotonic-GP-flow in both regression and preference tasks. Note that we do not compare against a GP with a M-spline because we showed in theorem 2 that our method provide the exact posterior for that model. Section 6.2 shows that the inclusion of monotonic constraints helps in a real world preferential learning problem. In particular, the logP score table shows that a SkewGPc provides much better logP score in the case of monotone options. Note that if the options do not have an underlying monotonic behaviour (other-options line) we do not improve over an unconstrained skewGP. Finally section 6.3 shows an application of preferential learning to desirability theory. In this case, we show again that a constrained skewGP performs better than unconstrained on monotonic options (logP table). Moreover, we show that the application of our method to this problem outperforms the state-of-the-art expected utility model.
>
> 3.  **random process is used to induce preferences, will its expressive ability be too weak**
> (Skew) Gaussian Processes are state-of-the-art models for preference learning. They learn the utility function underlying the preference relationship in a probabilistic manner by computing its posterior distribution. Their probabilistic nature, which allows them to model uncertainty, makes them particularly effective in modelling problems with small datasets. This is typical in many preference learning tasks where the datasets are small. Moreover, note that, the posterior mean of a SkewGP is comparable to the prediction of a Neural Network, but additionally SkewGPs provide a measure of uncertainty of their own predictions.
>
>
> 4. **Organization of the paper.** We regret that the reviewer found our paper to be  difficult to read. However, it's worth noting that other reviewers have found the paper to be clear and well-structured. However, we intend to address all reviewers' feedback in our revision process in order to improve its quality. Specifically, we will include  additional examples in  Section 4, a step-by-step algorithm in the Appendix describing how we compute the posterior under monotonicity constraints,  and provide clearer explanation of the objectives within each section of the numerical experiments.

---

### Official Review · Reviewer_czms · 2024-03-26

**Q2-1 Originality-Novelty:** 3
**Q2-2 Correctness-Technical Quality:** 3
**Q2-5 Clarity Of Writing:** 4

**Q1 Summary And Contributions:**

The paper focuses on the existing approaches to linearly constrained GP for regression. It can handle monotonic preference learning. The performance is demonstrated on simulated and real datasets.

**Q2-3 Extent To Which Claims Are Supported By Evidence:**

3: Good: the main claims are supported by convincing evidence (in the form of adequate experimental evaluation, proofs, (pseudo-)code, references, assumptions).

**Q2-4 Reproducibility:**

3: Good: key resources (e.g. proofs, code, data) are available and key details (e.g. proofs, experimental setup) are sufficiently well-described for competent researchers to confidently reproduce the main results.

**Q3 Main Strengths:**

The paper is well-written and follows naturally. The theorem and propositions are well stated.

**Q4 Main Weakness:**

I am not an expert in this field. Thus, I couldn't discuss more on the weakness. One possible concern is the limited novelty. It seems that the paper focuses on the application side.

**Q5 Detailed Comments To The Authors:**

N/A

**Q9 Complying With Reviewing Instructions:**

Yes

---

> ### Author Rebuttal · Authors · 2024-04-04
>
> We thank the reviewer for their positive comments. Regarding the novelty comment we would like to stress that, to the best of our knowledge, in the literature of Gaussian processes with monotonic constraints, the connection between operational points based methods and skewGP is new, which is a novel theoretical result. Indeed, this connection allows for an analytical calculation of the posterior distribution in both the regression and the preference case. This allows to naturally include monotonic constraints in preferential learning, which is also a novel theoretical result. Moreover, this allows us to save computational time, by exploiting the analytical derivations.
> We further refer the reviewer to the rebuttals of the other reviewers for more comments about this point.

---

### Official Review · Reviewer_3xyw · 2024-03-27

**Q2-1 Originality-Novelty:** 2
**Q2-2 Correctness-Technical Quality:** 3
**Q2-5 Clarity Of Writing:** 3

**Q1 Summary And Contributions:**

The aim is to learn strictly monotonic preferences from preference data by learning a strictly monotone utility function f. The paper gives a “SkewGP” approach. This turns out to be a Gaussian process with constraints at a finite set of locations.

**Q2-3 Extent To Which Claims Are Supported By Evidence:**

3: Good: the main claims are supported by convincing evidence (in the form of adequate experimental evaluation, proofs, (pseudo-)code, references, assumptions).

**Q2-4 Reproducibility:**

3: Good: key resources (e.g. proofs, code, data) are available and key details (e.g. proofs, experimental setup) are sufficiently well-described for competent researchers to confidently reproduce the main results.

**Q3 Main Strengths:**

1. A SkewGP prior and an affine-probit-normal product likelihood imply a SkewGP posterior. This seems nice and is from previous work. The contribution Theorem 1 is to extend this logic to impose monotonicity constraints at a finite number of locations called U.

2. It seems like there is no approximation error to using a finite U when using a finite kernel such as splines, which is, in my view, the strongest result.

3. The introduction explains the aims well.

**Q4 Main Weakness:**

1. The monotonicity constraint is only imposed at an apparently fixed, finite number of locations U called operational points when the kernel is infinite dimensional. It seems like an approximation error is being incurred since U is finite, but I did not see any discussion of this approximation error. How are U chosen? Is there a way to choose U that minimizes the approximation error?

2. “In the constrained case, however, the samples do not preserve monotonicity globally.” What does this mean? How should we interpret it?

3. It was not until page 3 that I appreciated that we observe regression-type observations and preference-type observations. In what contexts would we see a mix of observation types like this?

**Q5 Detailed Comments To The Authors:**

1. The notation t_1 and t_l is not easy to parse.

2. Please define xi following (1).

**Q9 Complying With Reviewing Instructions:**

Yes

---

> ### Author Rebuttal · Authors · 2024-04-06
>
> We thank the reviewer for their detailed comments. We answer below point by point to the reviewer's questions.
>
> Q1. There are many techniques to select the location of the operational points U, see Agrell (2019) for a review. In this work, we initialized the operational points with quantiles of the training inputs and then we treated them as model's hyper-parameters and optimized a variational lower bound. Note that the main focus of this work was to show that monotonic constraints can be implemented with a skewGP in a preferential learning setting. For this reason we did not study more complex methods for optimizing the operational points. We will make this point clearer in the revised version of the paper.
>
> Q2. **In the constrained case, however, the samples do not preserve monotonicity globally.** In the approach we discuss in Section 4.1, the constraints are only guaranteed to be satisfied at the operational points U. Previous works [49,58] have noticed that by selecting U appropriately (using the methods in Q1), it is possible to satisfy the constraints at all xs. Instead, for the approach in Section 4.2, the constraints are guaranteed to be satisfied at  all xs, no matter the location of the operational points. Those are the kernels we use in the numerical experiments section, therefore the constraints are guaranteed to hold globally.
>
> Q3. We would like to point out that our method can handle both regression-type and preference-type observation simultaneously, however it does not require them. For example in the numerical experiments in section 6, we use preference type data only in both experiments in section 6.2 and 6.3. In section 6.1 we use a benchmark function to generate two separate datasets: one for a regression task and the other one for a preference task. Note that the results shown in the table are related to different experiments.
>
> Q4. Our method can be applied to both mixed-type data but also to preference only data. Examples of mixed-type data can be found, for example, in Benavoli et al. (2021). A typical example (without monotonicity) is movies preferences: a user expresses preferences over movies by selecting movie A instead of movie B. Sometimes (but not always), the user will also rate the movie (3/5 stars). In this case, we have both preference and numeric (rating) data.
>
> We will clarify the notation $t_1, \ldots, t_\ell$ and define the vector $\xi \in \mathbb{R}^p$ in the revised version.
>
>
> Agrell, C. (2019). Gaussian Processes with Linear Operator Inequality Constraints. Journal of Machine Learning Research, 20(135), 1–36.
>
> Benavoli, A., Azzimonti, D., & Piga, D. (2021). A unified framework for closed-form nonparametric regression, classification, preference and mixed problems with Skew Gaussian Processes. Machine Learning, 110(11–12), 3095–3133.

---

### Meta-Review · Area_Chair_iPJM · 2024-04-16

The focus of this submission is preference learning, particularly learning strictly monotonic preferences from pairwise data, which under a continuity assumption boils down to finding a strictly monotone utility function representing it. Gaussian processes (GPs) represent a powerful method of learning functions; shape constraints (such as non-negative and monotonicity) are often imposed for them at finite many locations. The idea of the authors is that this strategy (a linearly constrained GP) can be interpreted as a skewGP (Theorem 1) which allows leveraging the conjugacy (Proposition 1) of skewGPs with affine-probit-normal product likelihood as specified in (4). This result is then translated to finite-dimensional kernels and transformed GPs to ensure monotonicity (Theorem 2-3) making use of M and I-splines. Both synthetic and real-world benchmarks (including preference learning and desirability learning) illustrate the practicality of the approach.

Kernels and Gaussian processes are among the most powerful techniques of our times. Enriching them with monotonicity constraints, and new tools allowing their computationally more effective implementation is of clear interest to the UAI community. The authors provide valuable insights in this direction.